# The effect of host community functional traits on plant disease risk varies along an elevational gradient

Fletcher W Halliday[1]*[†], Mikko Jalo[2][†], Anna-Liisa Laine[1,2]

[1]Department of Evolutionary Biology and Environmental Studies, University of Zürich, Zurich, Switzerland; [2]Faculty of Biological and Environmental sciences, University of Helsinki, Helsinki, Finland

**Abstract** Quantifying the relative impact of environmental conditions and host community structure on disease is one of the greatest challenges of the 21st century, as both climate and biodiversity are changing at unprecedented rates. Both increasing temperature and shifting host communities toward more fast-paced life-history strategies are predicted to increase disease, yet their independent and interactive effects on disease in natural communities remain unknown. Here, we address this challenge by surveying foliar disease symptoms in 220, 0.5 m-diameter herbaceous plant communities along a 1100-m elevational gradient. We find that increasing temperature associated with lower elevation can increase disease by (1) relaxing constraints on parasite growth and reproduction, (2) determining which host species are present in a given location, and (3) strengthening the positive effect of host community pace-of-life on disease. These results provide the first field evidence, under natural conditions, that environmental gradients can alter how host community structure affects disease.

*For correspondence:
Fletcher.w.halliday@gmail.com

[†]These authors contributed equally to this work

Competing interest: The authors declare that no competing interests exist.

## Introduction

Infectious disease is strongly influenced by host community structure and abiotic conditions (*Halliday et al., 2020a*; *Halliday et al., 2019*), both of which are undergoing unprecedented change as the climate is warming (*Pachauri et al., 2014*) and biodiversity is being reshuffled (*Díaz et al., 2019*; *Hillebrand et al., 2018*). Understanding how biotic and abiotic conditions interact to drive the emergence and spread of infectious disease is quickly emerging as one of the greatest research challenges of the 21st century and will be the key to limiting the impacts of infectious diseases on food production systems, wildlife, and humans. Disease ecology provides a framework for achieving this goal through careful examination of interactions among hosts, parasites, and the environment (*Johnson et al., 2015a*; *McNew, 1960*; *Seabloom et al., 2015*; *Figure 1a*). Yet, we have a poor understanding of how this framework operates under natural conditions, in part because several mechanisms can operate simultaneously, making it difficult to tease apart their relative contributions to realized disease risk.

Climate change involves increased environmental temperatures, which can profoundly alter disease risk (*Garrett et al., 2006*; *Harvell et al., 2002*; *Rohr et al., 2011*). These effects can result from direct impacts of environmental factors on parasite growth, survival and reproduction that underpin disease risk. For example, in an experiment in the Rocky mountains, host plants that grew on heated research plots showed increased disease, largely by increasing the amount of time that environmental conditions were favorable for parasite growth and reproduction (*Roy et al., 2004*). Importantly however, these same environmental factors can also indirectly influence disease risk by altering the composition of host or vector communities that are required for sustained parasite transmission (*Elad and Pertot, 2014*; *Garrett et al., 2006*; *Harvell et al., 2002*; *Mordecai et al.,*

**eLife digest** Climate change is causing shifts in the ecology and biodiversity of different world regions at unprecedented rates. Global warming is also linked with changes in the risk for certain infectious diseases in humans, but also in animals and plants. There are several possible mechanisms for this. For one thing, changing weather patterns may affect how pathogens grow and reproduce. For another, the distribution ranges of animal and plant hosts of certain disease-causing pathogens are changing because of global warming. This means that the distributions of pathogens are also changing, and so is the severity of the diseases that they cause.

Increasing temperatures may also influence the physiological traits that make host species suitable for pathogens. This is because the traits that allow species to survive or adapt to changes in their environment may also make them better at hosting and transmitting the pathogens that cause disease. For example, in plant communities, rising temperatures could favor species with faster growth rates, quicker reproduction and high dispersal, and these traits are often associated with more efficient spread of disease.

Despite a lot of research into the effects of climate, it remains unclear how temperature, pathogen growth and reproduction, and host species' traits and distributions combine and interact to alter infectious disease risk, especially in wild plant communities. To investigate this, Halliday, Jalo and Laine studied an area in southeast Switzerland where natural temperature and biodiversity change gradually through the region. The aim was to explore how relationships between plant biodiversity, pathogens and disease risk change with temperature, and to understand whether environmental or biological factors influence infectious disease risk more.

Halliday, Jalo and Laine measured the levels of fungal diseases found in the leaves of plant communities spanning 1,100 meters of elevation, showing that higher temperatures increase disease risk both directly and indirectly. Directly, higher temperatures increased pathogen growth and reproduction, and indirectly, they influenced which plants were present and therefore able to act as disease hosts. The results also indicated that temperature can affect how the traits of plants drive the transmission rates of fungal pathogens. Important predictors of disease risk were traits relating to the growth rate of host plants, which tended to increase in areas with low elevation where the surface of the soil was warm.

This study represents the first analysis, in wild plants, of how changing temperatures, the traits of shifting host species, and resident parasite populations interact to impact infectious disease risk. The insights Halliday, Jalo and Laine provided could aid in predicting how global climate change will influence infectious disease risk.

---

*2019*; *Newton et al., 2011*; *Yáñez-López, 2012*; *Rohr et al., 2011*), or by altering host defenses (*Descombes et al., 2017*; *Pellissier et al., 2018*; *Wolinska and King, 2009*). Thus, shifts in parasite replication that are driven by changing host or vector distributions can also determine whether and how changing environmental conditions will alter disease risk. There is growing empirical evidence in support of both direct effects that alter parasite growth and replication, as well as indirect effects that are mediated by changing host or vector community structure. However, disentangling the relative impacts of these direct and indirect effects of environmental factors on disease risk has been historically challenging, because it often requires a priori knowledge of environmental constraints acting on host and parasite populations (*Garrett et al., 2006*; *Harvell et al., 2002*; *Mordecai et al., 2019*; *Rohr et al., 2011*).

One way to disentangle direct and indirect effects of environmental conditions on disease is to consider these effects in the context of host functional traits. Host functional traits underlie ecologically important resource acquisition and allocation tradeoffs: hosts must balance allocating limited resources to maximize growth and reproduction, while also constructing tissue capable of withstanding stressful environmental conditions (*Díaz et al., 2016*; *Reich, 2014*; *Reich et al., 2003*; *Wright et al., 2004*). Thus, linking environmental conditions with relevant functional traits has become a tractable way to predict the richness and composition of communities (i.e. community structure) (*Cornelissen et al., 2003*; *Díaz and Cabido, 1997*; *Funk et al., 2017*; *Kattge et al., 2020*; *Lavorel and Garnier, 2002*; *McMahon et al., 2011*; *Reich, 2014*; *Sundqvist et al., 2013*).

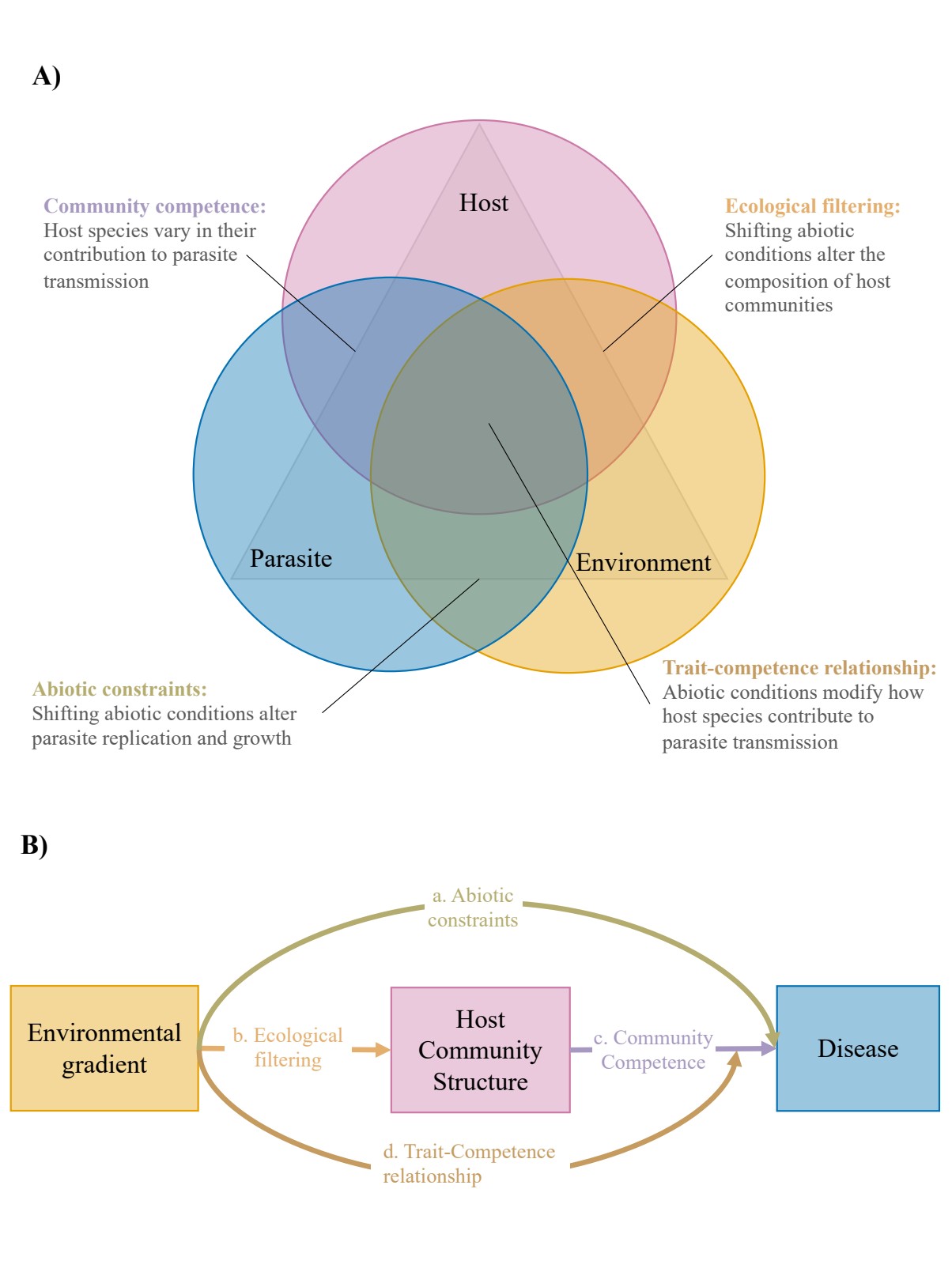

**Figure 1.** Relationships among hosts parasites and their environment at the scale of host communities. (**A**) The disease triangle (*McNew, 1960*) suggests that a combination of host, parasite, and environmental factors will influence whether disease is observed in a given location. Here, we conceptualize the disease triangle at the community level as consisting of three overlapping or interacting factors to demonstrate how the influence of environmental gradients on disease risk might depend on how these factors overlap. We highlight three potential processes that might occur in these

*Figure 1 continued on next page*

*Figure 1 continued*
areas of overlap, but acknowledge that other processes likely occur in these areas as well. (**B**) Conceptual metamodel of an environmental gradient directly influencing disease risk (path a), and indirectly influencing disease risk, both by altering host community structure (i.e. mediation; paths b and c), and by modifying how host community structure influences disease risk (i.e. moderation of the relationship between host traits and host competence, which we refer to as the trait-competence relationship; path d).

The functional traits expressed by those species that are able to colonize and persist in a given location can, in turn, affect disease risk (*Halliday et al., 2019*; *Johnson et al., 2013*; *Kirk et al., 2019*). Specifically, an infected host's ability to transmit disease to uninfected hosts, a trait often referred to as host competence, is often related to fast-growing, poorly defended tissues and short lifespans (*Becker and Han, 2021*; *Cronin et al., 2014*; *Cronin et al., 2010*; *Huang et al., 2013*; *Johnson et al., 2012*; *Martin et al., 2019*; *Martin et al., 2016*; *Parker and Gilbert, 2018*; *Stewart Merrill and Johnson, 2020*; *Welsh et al., 2020*). Importantly, these functional trait values also underlie ecological tradeoffs related to host growth and defense, resource acquisition and allocation, and survival and reproduction (i.e. life history) (*Coley et al., 1985*; *Herms and Mattson, 1992*; *Martin et al., 2016*; *Reich, 2014*; *Reich et al., 2003*; *Ricklefs and Wikelski, 2002*; *Stearns, 1992*; *Stearns, 1989*; *Wright et al., 2004*). Thus, host community competence (a community-level metric of host competence) is expected to correspond to the same functional traits (i.e. host pace-of-life) that link host community structure to shifting environmental conditions.

A trait-based framework of host community competence may explain why biodiversity loss is consistently associated with higher disease risk (*Halliday et al., 2020b*; *Johnson et al., 2013*; *LoGiudice et al., 2003*; *Ostfeld and LoGiudice, 2003*), a relationship known as the 'dilution effect' of biodiversity (*Keesing et al., 2010*; *Keesing et al., 2006*; *Ostfeld and Keesing, 2000*). This is because host species that are most resistant to biodiversity loss or best able to colonize newly disturbed habitats often rely on the same life-history strategies that are associated with higher host competence (*Johnson et al., 2013*; *LoGiudice et al., 2003*; *Ostfeld and LoGiudice, 2003*). For example, species that are associated with habitat fragmentation, a key anthropogenic driver of biodiversity loss, are often characterized by life history strategies favoring a 'fast pace-of-life' (i.e. fast growth rates, quick reproduction, and high dispersal) (*Albrecht and Haider, 2013*; *Fay et al., 2015*; *Gibbs and van Dyck, 2010*; *Hanski et al., 2006*; *Keinath et al., 2017*; *Merckx et al., 2018*; *Ziv and Davidowitz, 2019*). But this fast pace-of-life often comes at the cost of reduced defense against parasites (*Cappelli et al., 2020*; *Coley et al., 1985*; *Cronin et al., 2014*; *Cronin et al., 2010*; *Heckman et al., 2019*; *Herms and Mattson, 1992*; *Johnson et al., 2012*; *Sears et al., 2015*). Thus, habitat fragmentation can increase disease by increasing the density of fast pace-of-life, highly competent hosts, while slow pace-of-life, less-competent hosts are lost (*Johnson et al., 2015b*; *Joseph et al., 2013*; *Mihaljevic et al., 2014*). This hypothesis has widespread empirical support in a variety of systems (*Johnson et al., 2019*; *Johnson et al., 2013*; *Liu et al., 2018*; *Ostfeld and LoGiudice, 2003*). Shifting community structure during biodiversity loss may therefore predictably influence infectious disease risk (*Halliday et al., 2020b*).

Although relationships between host community structure and disease risk are becoming increasingly appreciated, how these relationships change across environmental gradients remain unknown (*Halliday et al., 2020b*; *Halliday and Rohr, 2019*). The relationship between host traits and host competence can be variable, and this relationship might also depend on the environmental context in which host-parasite interactions play out (*Figure 1b.*, path d). For example, *Welsh et al., 2016* showed that when hosts were reared under novel resource conditions, trait-based models of host susceptibility became increasingly inaccurate, because novel resource conditions altered how traits covaried with one another and how raw trait values predicted infection. Thus, traits associated with host community competence in one environment might not predict host community competence across environmental gradients.

We hypothesized that three non-mutually exclusive mechanisms would determine how environmental conditions influence disease risk in host communities: (1) *directly, by altering parasite growth and reproduction* (i.e. through abiotic constraints; *Figure 1b.*, path a), (2) *indirectly, by altering which host species occur in which locations* (i.e. mediated by shifting host community structure; *Figure 1b.*, paths b and c), and (3) *indirectly, by altering how host traits influence parasite transmission* (i.e.

moderated by altering the relationship between host traits and host competence, which we refer to as the trait-competence relationship; *Figure 1b.*, path d).

Here, we test the relative contributions of these three mechanisms through which environmental conditions can drive infectious disease risk (i.e. direct, mediated, and moderated) by measuring foliar fungal disease in host plant communities along a roughly 1100 m elevational gradient in Southeastern Switzerland. Foliar fungal parasites are a widely used, tractable model of disease risk that respond to small-scale variation in host community structure and environmental conditions (*Cappelli et al., 2020*; *Halliday et al., 2019*, *Halliday et al., 2017*; *Liu et al., 2018*; *Liu et al., 2017*; *Liu et al., 2016*; *Mitchell et al., 2003*; *Mitchell et al., 2002*; *Rottstock et al., 2014*). Host community structure and environmental conditions, in turn, vary predictably with elevation (*Grinnell, 1914*; *Halbritter et al., 2018*; *Malhi et al., 2010*; *Sundqvist et al., 2013*; *Whittaker, 1956*). Thus, an elevational gradient represents a natural laboratory for studying long-term, large scale changes in climate as well as interacting biotic and abiotic factors that are associated with climate change (*Alexander et al., 2015*; *Fukami and Wardle, 2005*; *Sundqvist et al., 2013*).

Our study reveals strong evidence that increasing temperature associated with lower elevation can directly influence disease risk, which we attribute to well-established effects of abiotic conditions (*Avenot et al., 2017*; *Garrett et al., 2006*; *Harvell et al., 2002*; *Tapsoba and Wilson, 1997*; *Waugh et al., 2003*) on parasite replication and growth, and can indirectly influence disease risk by shifting host community structure and by modifying the trait-competence relationship. Together, these results highlight the need to consider biotic and abiotic drivers jointly, in order to predict disease risk in the face of climate change.

## Results

To evaluate abiotic constraints on parasite replication and growth (i.e. direct effects), shifting host community structure (i.e. mediation; *Baron and Kenny, 1986*), and modification of the trait-competence relationship (i.e. moderation; *Baron and Kenny, 1986*) as mechanisms through which environmental gradients can influence disease risk, we surveyed 220, 0.5 m-diameter vegetation communities (i.e. small plots), that were established in four meadows along a 1101 m elevational gradient as part of the Calanda Biodiversity Observatory (CBO) in 2019 in order to investigate biotic and abiotic drivers of species interactions (*Figure 2*; *Supplementary file 1a*).

### Association between elevation and environmental factors

The elevational gradient captured by the CBO allows us to explore associations among abiotic factors and biodiversity while minimizing other confounding factors like day length, geology, and biogeographic history (*Halbritter et al., 2018*). We assessed the association between elevation and environmental conditions by fitting linear models. Mean soil, soil surface, and air temperature strongly and consistently decreased with increasing elevation ($p < 0.001$, $R^2 = 0.88$; $p < 0.001$, $R^2 = 0.86$; $p < 0.001$, $R^2 = 0.89$; respectively), while mean soil moisture was uncorrelated with elevation ($p = 0.72$, $R^2 = 0.006$). The mean soil surface temperature at sites located in the highest elevation meadow (1576 m–1749 m) was, on average, 4.67 °C lower than sites located in the lowest elevation meadow (648 m–766 m). The altitudinal temperature lapse rate along the elevational gradient was –0.57 °C/100 m.

### Effect of environmental conditions on host community structure

In total, 188 host taxa were observed across the 220 small plots of the CBO. The communities consisted mostly of perennial herbs such as *Salvia pratensis* and *Helianthemum nummularium*, and were dominated by grasses that tolerate grazing such as *Dactylis glomerata*, *Lolium perenne,* and *Phleum pratense*. The most abundant species was *Brachypodium pinnatum*. An herbarium specimen of each taxon encountered is deposited at the University of Zürich. We assessed the relationship between abiotic conditions and species richness by fitting linear mixed models with large plots, sites, and meadows as nested random intercepts. Species richness in the small plots varied from 7 to 30 species (median 20), was uncorrelated with soil moisture ($p = 0.98$) and increased as elevation increased and soil surface temperature declined ($p = 0.005$; Marginal $R^2 = 0.10$; Conditional $R^2 = 0.75$), with median species richness roughly 16% higher in plots located at the highest elevation meadow, characterized by the coolest environmental temperatures, compared to the lowest elevation meadow,

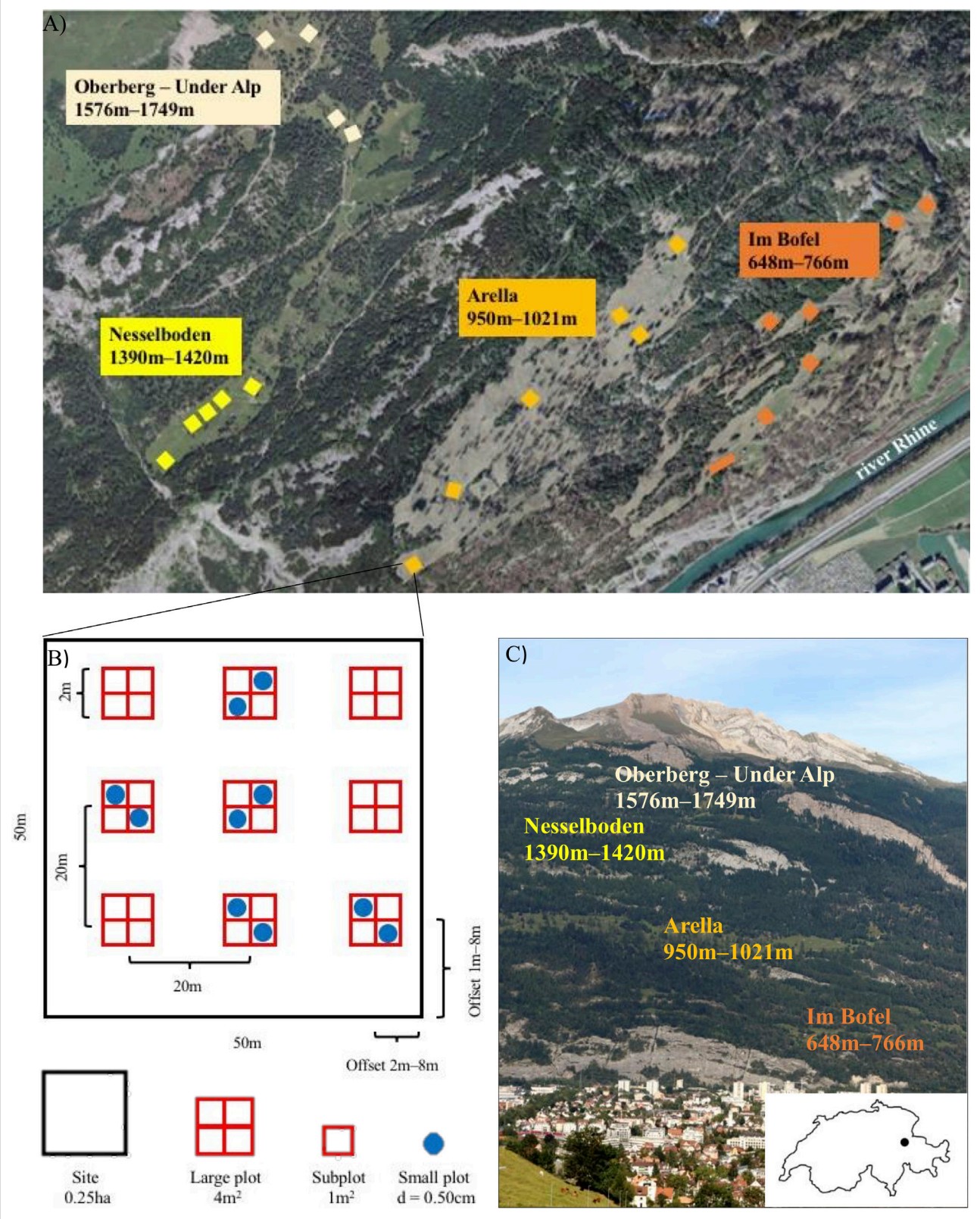

**Figure 2.** Overview of the Calanda Biodiverstity Observatory. (**A**) Study meadows and sites on mount Calanda. Photo: Federal Office of Topography SwissTopo 2020, editing: Mikko Jalo (**B**) Example of the arrangement of large and small plots within a site. (**C**) The study meadows on mount Calanda. Photo and editing: Mikko Jalo.

*Figure 2 continued on next page*

*Figure 2 continued*

The online version of this article includes the following figure supplement(s) for figure 2:

**Figure supplement 1.** Images representing survey methods.

which was characterized by the warmest environmental temperatures (*Supplementary file 1b*). These effects were qualitatively similar when we included air temperature and elevation in place of soil-surface temperature, though the relationship became nonsignificant when we replaced soil-surface temperature with soil temperature in the model (p = 0.15; *Supplementary file 1b*).

We performed confirmatory factor analysis to assign six foliar functional traits associated with the worldwide leaf economics spectrum to a single axis representing host pace-of-life. One trait, photosynthetic rate, loaded particularly poorly on this axis (factor loading 0.036), and was therefore excluded from the latent factor. This resulted in a single factor, explaining 62% of the variance in specific leaf area, 51% of the variance in leaf chlorophyll content, 25% of the variance in leaf nitrogen, 10% of the variance in leaf phosphorus, and 2% of the variance in leaf lifespan ($\chi^2$ (df = 5) = 4.24, p = 0.52; CFI = 1.019; *Figure 3—figure supplement 1*). Consistent with resource-acquisition and allocation tradeoffs (*Díaz et al., 2016*; *Reich, 2014*; *Wright et al., 2004*), higher values of host pace-of-life were associated with increases in specific leaf area, leaf chlorophyll content, leaf nitrogen, and leaf phosphorus, and with shorter leaf lifespans. We then used each species' unique score on this pace-of-life factor to quantify the community-weighted mean host pace-of-life (hereafter community pace-of-life) for each small plot. We assessed the relationship between abiotic conditions and community pace-of-life by fitting linear mixed models with large plots, sites, and meadows as nested random intercepts. Although host community pace-of-life was unrelated to soil moisture (p = 0.13), host community pace-of-life declined with reduced soil-surface temperature associated with higher elevation (p = 0.010; Marginal $R^2$ = 0.11; Conditional $R^2$ = 0.83; *Supplementary file 1b*; *Figure 3—figure supplement 2*), consistent with expectations regarding shifting host community structure (*Descombes et al., 2017*; *Hulshof et al., 2013*; but see *Pellissier et al., 2018*). These effects were qualitatively similar when we included soil temperature or air temperature in place of soil-surface temperature in the model, though the effect became marginally nonsignificant when we replaced temperature with elevation in the model (p = 0.066; *Supplementary file 1b*; *Figure 3—figure supplement 2*).

## Model testing effects of environmental conditions, community structure, and their interaction on disease

We tested whether the relationship between host community structure (i.e. host species richness and host community pace-of-life) and disease would change as a function of environmental conditions by fitting a linear mixed model with square-root transformed community parasite load (e.g. *Halliday et al., 2019*, *Halliday et al., 2017*; *Mitchell et al., 2002*) as the response. Soil-surface temperature, soil moisture, host community species richness, pace-of-life, and pairwise interactions between both measures of community structure and each abiotic variable were treated as fixed effects, with large plots, sites, and meadows as nested random intercepts. All variables that were treated as fixed

**Table 1.** Results of type II analysis of deviance test on the mixed model of disease, testing whether each factor influenced square-root transformed community parasite load.

| Predictor | Estimate | Chisq | Df | P |
|---|---|---|---|---|
| Soil-surface Temperature | 0.044 | 7.4236 | 1 | 0.0064 |
| Soil Moisture | −0.254 | 0.1390 | 1 | 0.7092 |
| Host Richness | −0.009 | 5.3325 | 1 | 0.0209 |
| Host Pace-of-Life | 0.133 | 1.6970 | 1 | 0.1926 |
| Temperature × Richness | 0.004 | 2.6551 | 1 | 0.1032 |
| Temperature × Pace-of-Life | 0.118 | 11.2498 | 1 | 0.0008 |
| Moisture × Richness | −0.288 | 2.7677 | 1 | 0.0962 |
| Moisture × Pace-of-Life | −2.037 | 0.5647 | 1 | 0.4524 |

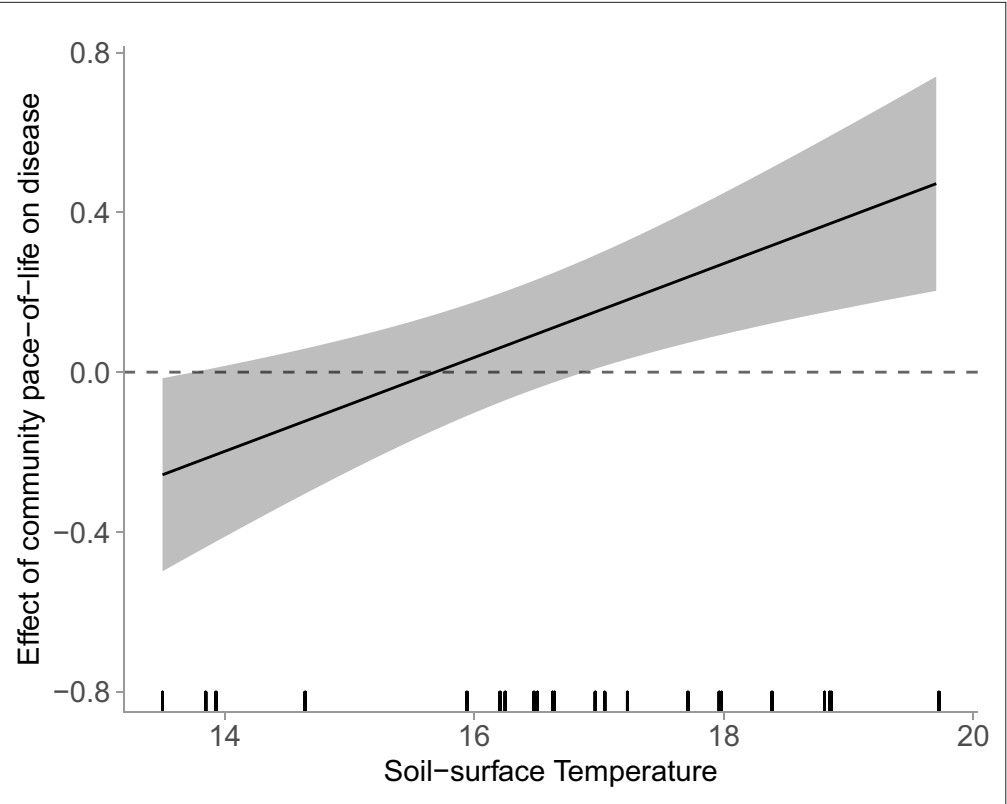

**Figure 3.** Effect of host community pace-of-life on disease as a function of increasing soil-surface temperature. Model estimated effects of soil-surface temperature on the slope of the relationship between host community pace-of-life and (square-root-transformed) parasite community load (i.e. the interactive effect of host community pace-of-life and soil-surface temperature on disease, which represents a changing trait-competence relationship), estimated from the raw (i.e. unstandardized) coefficients of the linear mixed model testing effects of environmental conditions, community structure, and their interaction on disease. The rug along the *x*-axis shows the distribution of the empirical data. Communities that experience the highest soil-surface temperatures (i.e. located at the lowest elevation) exhibit the strongest positive relationship between host pace-of-life and disease. That positive relationship weakens as temperature declines, and below mean-soil surface temperatures of 17.5 C (i.e. above 1000 m), there is no relationship between host pace-of-life and disease.

The online version of this article includes the following figure supplement(s) for figure 3:

**Figure supplement 1.** Host taxa arranged by their mean absolute vegetative cover in plots where those taxa occur (y axis) and host pace-of-life (x-axis).

**Figure supplement 2.** Relationship between host richness, host community pace-of-life (together measuring host community structure), soil-surface temperature, soil temperature, air temperature, and elevation.

effects in the model were centered so that the mean value of each variable was used as the reference value for interpreting the other variables' independent effects. This mixed model of disease revealed several independent and interactive effects of host community structure and environmental conditions on disease risk (Marginal $R^2$ = 0.227; Conditional $R^2$ = 0.497; RMSE = 0.292; LOOCV RMSE = 0.311; *Table 1*). Consistent with the hypothesis that host pace-of-life can determine host community competence, communities that were dominated by hosts with fast-paced life-history strategies exhibited the most disease, but this effect declined as elevation increased and temperature declined (temperature × pace-of-life: p < 0.001). This weakening effect of host community pace-of-life as soil-surface temperature declined is consistent with the hypothesis that abiotic conditions can alter which traits favor parasite transmission through the relationship between host competence and disease risk (*Figure 3*). These results therefore provide field evidence that an environmental gradient can alter the effect of host community structure on disease risk.

The model also revealed significant independent effects of host community structure and abiotic conditions on disease risk. Specifically, the model revealed evidence supporting the dilution effect

hypothesis: increasing species richness was associated with a reduction in disease (p = 0.021), and this effect was independent of soil-surface temperature (temperature × richness: p = 0.10). Community parasite load was also positively associated with increasing soil-surface temperature (p = 0.006), consistent with the hypothesis that environmental gradients can alter parasite growth and reproduction via abiotic constraints. These effects were qualitatively similar when we included soil temperature, air temperature, or elevation in place of soil-surface temperature in the model (*Supplementary file 1c*).

In contrast with results involving soil-surface temperature, there was no statistically significant linear relationship between soil moisture and disease (p = 0.71), nor was there a significant interaction between soil moisture and host richness (p = 0.10) or community-weighted mean pace-of-life (p = 0.45) on disease. Because soil moisture was unrelated to elevation, pace-of-life, species richness, and disease in our models, this factor was omitted from further analyses.

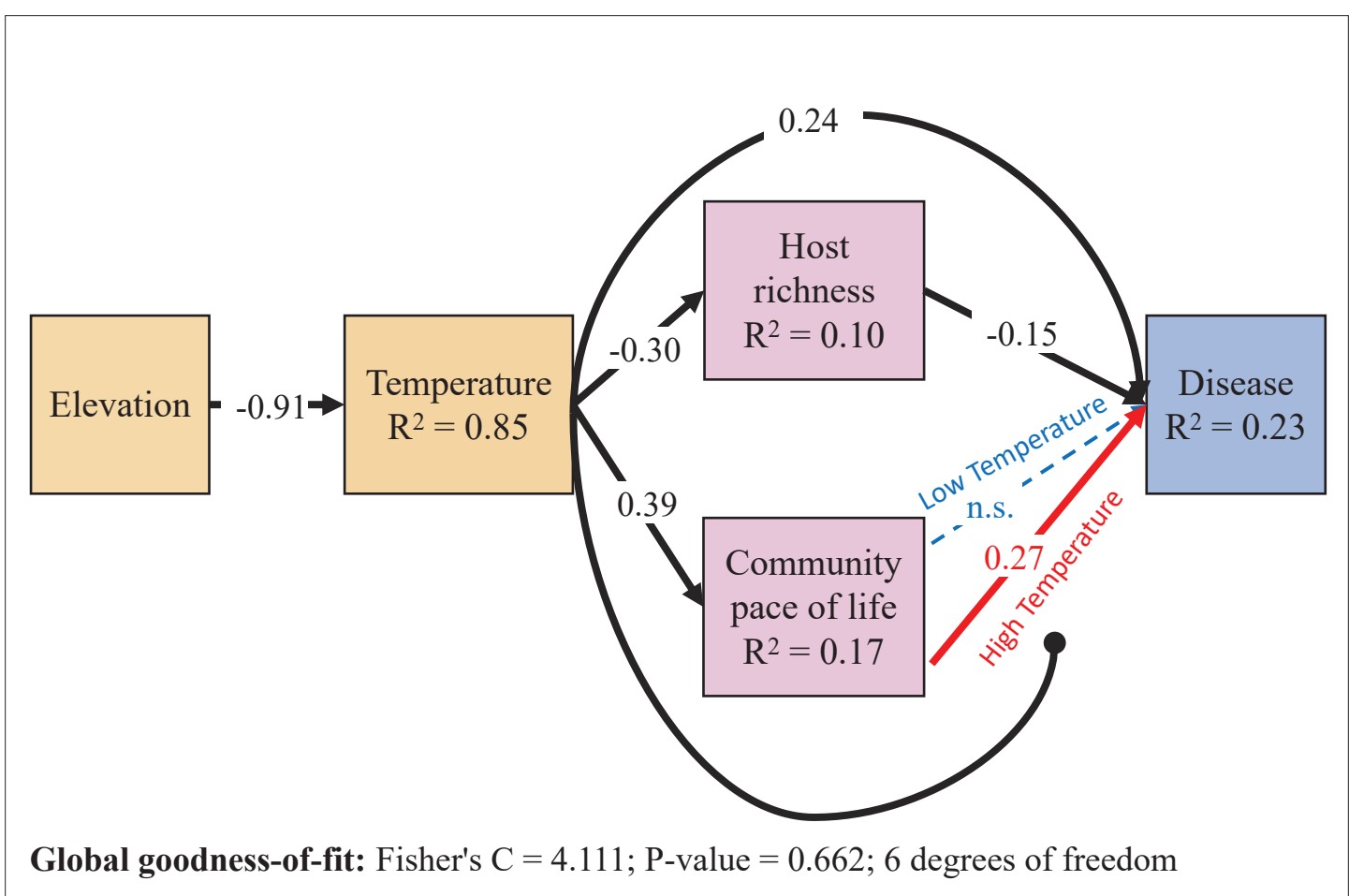

**Global goodness-of-fit:** Fisher's C = 4.111; P-value = 0.662; 6 degrees of freedom

**Figure 4.** Results from the piecewise structural equation model. Dashed lines are not supported by the model (p > .05). All coefficients are scaled by the ratio of the standard deviation of x divided by the standard deviation of y (i.e. standardized estimates), and therefore differ from the values in *Figure 3*. Correlations between errors were not supported by the model and are not shown. Colors are drawn to highlight the statistical interaction between host community pace-of-life and temperature. The High and Low Temperature coefficients are estimated with the reference temperature set to one standard deviation above and below the mean temperature, respectively. All other coefficients are estimated from a model using mean-centered values for temperature and community pace-of-life. Higher soil-surface temperature, associated with lower elevation, increased disease through three non-mutually exclusive pathways: directly via abiotic constraints, and indirectly both via shifting host community structure as well as by altering the trait-competence relationship.

## Structural equation model comparing direct and indirect effects of environmental conditions on disease

Together, models of host community species richness and pace-of-life showed that declining temperature associated with increasing elevation could determine changes in host community structure, and the model of disease showed that host community structure and temperature could independently and interactively influence disease. To explore the relative influence of these direct and indirect effects on disease risk, we next constructed a structural equation model. Our data were well fit by this model (Fisher's C = 4.111; p-value = 0.662; 6 degrees of freedom, *Supplementary file 1d*; *Figure 4*). The model leverages the strong, negative effect of elevation on soil-surface temperature (standardized path coefficient = –0.91, $R^2$ = 0.85) to compare three separate pathways through which increasing temperature can increase disease: First, increasing temperature increased community parasite load directly (standardized path coefficient = 0.24). Second, increasing temperature increased community parasite load indirectly by reducing host species richness (i.e. via mediation; product of standardized path coefficients = 0.045). Third, increasing temperature increased community parasite load indirectly by simultaneously increasing host community pace-of-life (i.e. via mediation; mean-centered standardized path coefficient = 0.39) and strengthening the relationship between host pace-of-life and disease risk (i.e. via moderation; mean-centered standardized path coefficient = 0.18; *Figure 4*). Together these results highlight the pressing need to consider host community context in predicting how shifting environmental gradients will alter disease risk.

## Discussion

This study shows, to our knowledge, the first evidence under natural field conditions that, in addition to directly influencing disease risk, the abiotic environment can also indirectly influence disease both by altering host community structure (i.e. mediation) and by modifying how host community structure influences disease risk (i.e. moderation). Furthermore, this linkage between the abiotic environment and host community structure suggests that any single factor would be inadequate for explaining disease risk along our environmental gradient. Together, these results reveal the role that host communities play in determining ecosystem health across environmental gradients, suggesting that predicting how shifting abiotic conditions will influence disease risk will require explicit consideration of how host and parasite communities jointly respond to the abiotic environment.

Our results indicate that increasing temperatures associated with lower elevation, can independently influence disease. Specifically, increasing temperature increased disease, even after accounting for effects of host community structure on disease. We hypothesize that reduced temperature associated with increasing elevation may have reduced disease directly by constraining parasite growth, survival, and reproduction. Many foliar parasites grow and reproduce more successfully in warmer environmental temperatures (*Avenot et al., 2017*; *Garrett et al., 2006*; *Harvell et al., 2002*; *Tapsoba and Wilson, 1997*; *Waugh et al., 2003*). Warmer temperatures can also increase parasite overwintering success (*Burdon and Elmqvist, 1996*; *Pfender and Vollmer, 1999*) or allow parasites to produce more generations during a longer growing season (*Garrett et al., 2006*). These results corroborate past studies suggesting that environmental gradients can directly alter the strength of biotic interactions (*Descombes et al., 2017*; *Hargreaves et al., 2019*; *Pellissier et al., 2014*; *Roslin et al., 2017*; *Schemske et al., 2009*), including host-parasite interactions (*Abbate and Antonovics, 2014*; *Allen et al., 2020*; *LaManna et al., 2017*; *Nunn et al., 2005*). However, despite the strong and consistent effect of increasing temperature on disease, temperature was highly correlated with elevation, and we cannot rule out the possibility that these effects might be driven by some other factor associated with elevation that was not measured, such as changing humidity or soil nutrient availability. Thus, temperature effects should be interpreted with some caution.

In addition to directly influencing disease, our results indicate that increasing temperature can also indirectly influence disease by altering host community structure. Specifically, increasing temperature reduced host species richness, which, in turn, reduced disease. The reduction in host species richness with increasing temperature might be attributable to the occurrence of both low-elevation and high-elevation adapted species occupying the coolest study sites, located at the highest elevation (*Colwell and Lees, 2000*). Communities in the highest elevation meadow, located just below the tree line, included plant species characteristic of low elevations (e.g. *Lathyrus pratensis*, *Lolium perenne*, and

*Salvia pratensis*) and plant species that tend to occupy high elevation grasslands (*Soldanella alpina*, *Ranunculus montanus,* and *Carex sempervirens*), indicating that these high-elevation sites represent an intermediate zone between subalpine and alpine vegetation communities.

Host communities with higher species richness, in turn, experienced less disease (i.e. a dilution effect; *Keesing et al., 2010*; *Keesing et al., 2006*), even after accounting for the direct effects of temperature on disease and other measures of host community structure. Past studies indicate that increasing biodiversity is often associated with a decline in disease risk because host community structure shifts during biodiversity loss to favor more competent hosts (*Johnson et al., 2019*; *Joseph et al., 2013*; *Liu et al., 2018*; *LoGiudice et al., 2003*; *Ostfeld and LoGiudice, 2003*; *Rohr et al., 2020*). However, in contrast with past studies focused on biodiversity loss, our study measured biodiversity change across a natural biodiversity gradient, which is not expected to consistently influence disease risk (*Halliday et al., 2020b*). We hypothesize that increasing species richness may have reduced disease risk in this system by reducing host density (*Keesing et al., 2006*; i.e. via encounter reduction; *Mitchell et al., 2002*). Encounter reduction might be particularly relevant in this system, because, in addition to altering host richness, reduced temperatures associated with increasing elevation also influence the length and timing of the growing season, which can affect peak prevalence and the duration of the epidemic season.

In addition to direct and indirect effects via mediation, our results further indicate that increasing temperature can indirectly influence disease by modifying the effect of host community structure on disease (i.e. via moderation). Specifically, disease was influenced by host community pace-of-life, but only at high temperature, low elevation sites. Because more competent hosts often exhibit fast-paced life history strategies (*Cronin et al., 2010*; *Johnson et al., 2012*; *Martin et al., 2016*; *Parker and Gilbert, 2018*; *Welsh et al., 2020*), we expected that host communities dominated by species with a fast pace-of-life would experience greater disease. However, a prior study suggested that the relationship between host traits and host competence might be sensitive to environmental conditions (*Welsh et al., 2016*), which we hypothesized would cause the relationship between host community pace-of-life and disease risk to shift across environmental gradients. Our analysis was consistent with this hypothesis: increasing temperature not only modified host community pace-of-life, but the effect of host community pace-of-life on disease was also sensitive to increasing temperature. Host community pace-of-life most strongly predicted disease risk at the highest temperatures, associated with the lowest elevation, but this effect weakened and ultimately disappeared as elevation increased and temperature declined.

These results indicate that warming temperatures can modify the effect of host community pace-of-life on disease risk, which we attribute to a change in the relationship between host traits and host competence across environmental conditions. However, we cannot rule out the possibility that the interaction between host pace-of-life and temperature could have also been driven by other mechanisms. For example, the values of functional traits expressed by a single species may have changed along the environmental gradient via a phenomenon known as intraspecific trait variation (*Albert et al., 2011*; *Funk et al., 2017*; *Messier et al., 2010*; *Violle et al., 2012*). Studies of functional traits (including this study) typically characterize each species with a single value for each trait, such as the species-level mean, under the assumption that ecologically important traits vary more among species than within species (*McGill et al., 2006*). However, functional traits of individuals within a species can vary due to local adaptation and phenotypic plasticity driven by local context (*Albert et al., 2011*; *Funk et al., 2017*; *Messier et al., 2010*; *Violle et al., 2012*). Thus, intraspecific shifts in the expression of key functional traits across our elevational gradient could drive the apparent interaction between host community pace-of-life and temperature. Alternatively, a reduction in infection severity with cooling temperatures could weaken the importance of investment in disease resistance (*Benkman, 2013*; *Thompson, 1999*). Thus, host species may still form strong trade-offs in fast vs slow strategies for growth vs. survival, but this pace-of-life trait would have a weak link with disease severity. Future studies should explore these mechanisms by directly measuring host and parasite functional traits across environmental gradients like elevation.

Together, the results of this study highlight the need to consider host community context in predicting how climate change will alter disease risk. Specifically, in this study, effects of the abiotic environment and changing environmental temperature on disease strongly depended on shifting host community pace-of-life. These results are consistent with a growing body of literature suggesting

that the role of host communities in regulating ecosystem processes is at least partially explained by characteristics of species present in those ecosystems (*Allan et al., 2015*; *Heilpern et al., 2020*; *Le Bagousse-Pinguet et al., 2019*; *Leitão et al., 2016*; *Mouillot et al., 2011*; *Start and Gilbert, 2019*; *Van de Peer et al., 2018*), but that abiotic factors such as temperature can override the effects of biotic factors on ecosystem processes (*Cannone et al., 2007*; *Laiolo et al., 2018*). These results therefore suggest that predicting how climate change will influence disease may depend on complex relationships between environmental factors and the structure of host communities.

## Materials and methods

### Study system

The Calanda Biodiversity Observatory (CBO) consists of four publicly owned meadows located along a 1101 m elevational gradient (648 m to 1749 m) below tree-line on the south-eastern slope of Mount Calanda (46°53′59.5″N 9°28′02.5″E) in the canton of Graubünden (*Figure 2*). The mean annual temperature at 550 m altitude is 10°C and the mean annual precipitation is 849 mm (*MeteoSwiss, 2020*), with temperature declining and precipitation increasing as elevation increases (e.g. in 2013 and 2014, mean temp and precipitation at 1400 m were 7°C and 1169 mm, respectively; *Alexander et al., 2015*). The soil in the area is generally calcareous and has low water retention (*Alexander et al., 2015*; *Eggenberg and Möhl, 2013*). The four CBO meadows are variable in size (roughly 8–40 Ha), and separated by forests and at least 500 m elevation. Meadows are maintained through grazing and mowing, a typical form of land use in the Swiss Alps (*Bätzing, 2015*), and cover collinean (< 800 m) mountain (800 m–1500 m) and subalpine (1500–2200 m) vegetation zones (*Eggenberg and Möhl, 2013*; *Ozenda, 1985*). The CBO meadows are grazed by cattle twice per year as the cattle are moved between low and high altitudes.

Increasing elevation is associated with changes in a variety of abiotic conditions, including a reduction in temperature. Temperature decreases approximately 0.4–0.7 °C for each 100 m increase in elevation because of lower air pressure in high elevations, a phenomenon known as the altitudinal temperature lapse rate (*Barry, 2008*). The altitudinal temperature lapse rate varies among years and even days, usually being lower in winters and during nights. Typical altitudinal temperature lapse rates in the Alps vary from –0.54°C/100 m to –0.58°C/100 m (*Rolland, 2003*).

### Study design

The CBO consists of a nested set of observational units (*Figure 2*). Each meadow contains 4–7, .25 ha sites (n = 22 sites). Sites were selected to maximize coverage over each meadow, avoiding roads that would cross the sites and large trees, shrubs and rocks that could create a forest- or shrub-type habitat that differs from grassland, and were placed sufficiently far from forest edges so that they were not shaded by the forest canopy. Each site is 50 m x 50 m and contains a grid of nine evenly spaced, 4 m$^2$ large-plots, with the exception of one site (I3), which is 100 m x 25 m and contains 10 large plots due to its shape. Altogether, there are 199 large plots. In each site, large plots are arranged in a grid with the center of each plot separated by at least 20 m distance from its nearest neighbor. The location of the grid was randomized within each site and always located at least 2 m from the site edge. Each large plot is subdivided into four, 1 m$^2$ subplots (n = 796). At each site, five large plots were selected to contain an intensively surveyed module (ISM), which consisted of two 50 cm-diameter, round small plots, placed in opposite subplots (n = 110 ISMs consisting of 220 small plots). These intensively surveyed small plots are the smallest unit of observation used in this study (*Figure 2*).

### Quantification of host community structure

In July 2019, we recorded the identity and visually quantified the percent cover of all plant taxa in each small plot (n = 220). Vegetation surveys entailed the same two researchers searching within the subplot area for all vascular plants present in the subplot, before jointly estimating the total percent cover of each species (*Halbritter et al., 2020*). Plant individuals that were growing outside the small plot, but whose foliage extended into the small plot, were included in this survey. Plant taxa were identified with the help of plant identification literature (*Eggenberg et al., 2018*; *Eggenberg and Möhl, 2013*; *Lauber et al., 2018*). The survey started at the lowest elevation and continued higher in order to survey the meadows approximately at the same phase of the growing season in relation to

one another. The survey was initiated at least 4 days after cows had grazed each meadow (*Supplementary file 1a*).

We evaluated changes in two components of host community structure to evaluate indirect effects of environmental conditions on disease: host species richness and community-weighted mean host pace-of-life. These two components of host community structure commonly respond to changing environmental conditions (*Descombes et al., 2017*; *Hulshof et al., 2013*), and represent important characteristics of host communities that influence disease risk (*Joseph et al., 2013*; *Liu et al., 2018*; *Liu et al., 2017*). We quantified community-weighted mean host pace-of-life using the TRY database (*Kattge et al., 2020*). We first extracted six traits for every host taxon in the database (plant photosynthetic rate, leaf chlorophyll content, leaf lifespan, leaf nitrogen content, leaf phosphorus content, and specific leaf area), omitting tree seedlings, which are functionally dissimilar from the more dominant herbaceous taxa, and taxa that could not be identified to host genus, which together, never accounted for more than 7% cover in a plot (mean = 0.04%). Unknown taxa that could be identified to the genus level were assigned genus-level estimates for each host trait, by taking the mean of the trait value for all members of that genus that had been observed on Mount Calanda during extensive vegetation surveys (*Supplementary file 1e*). We then performed full-information maximum-likelihood factor analysis to produce a single axis representing covariation in the functional traits associated with host pace-of-life using the umxEFA function in r-package umx (*Bates et al., 2019*). This approach allows each host taxon to be assigned a value for host pace-of-life, even if that taxon is missing some values for individual functional traits. Finally, we calculated a single value for each small plot (n = 220) using the community-weighted mean of host pace-of-life (hereafter community pace-of-life). The community weighted mean (CWM) was calculated as:

$$CWM = \sum_{i=1}^{Nsp} p_i x_i$$

where Nsp is the number of taxa within a plot with a pace-of-life trait value in the dataset, $p_i$ is the relative abundance of taxon, *i*, in the plot (i.e. the absolute vegetative cover of taxon, *i*, divided by the total absolute cover of all taxa in the plot), and $x_i$ is the host pace-of-life value for taxon, *i*.

## Quantification of disease

A survey of foliar disease symptoms was carried out in August 2019 by estimating the percent of leaf area damaged by foliar fungal parasites on up to five leaves of twenty randomly selected host individuals per small plot (n = 18,203 leaves on 4400 host individuals across 220 small plots). The disease survey was conducted by placing a grid of 20 equally spaced grill sticks into the ground, with each stick having a distance of 10 cm to its nearest neighbor (*Figure 2—figure supplement 1*). The 20 plant individuals that were most touching the sticks were then identified, and the five oldest non-senescing leaves on each plant were visually surveyed for foliar disease symptoms following the plant pathogen and invertebrate herbivory protocol in *Halbritter et al., 2020*. The survey was carried out on leaves, because symptoms are highly visible and easily grouped into parasite types on leaves. On each leaf, we estimated the leaf area (%) that was covered by disease symptoms. Some plant individuals had fewer than five leaves, so fewer than five leaves were surveyed on those plants. Unlike the vegetation survey, the disease survey was not conducted in elevational order due to logistical constrains related to site accessibility. Small plots were surveyed between 29 July and 19 August 2019 (*Supplementary file 1a*), which we observed to be time of peak plant biomass in this system.

Disease was assessed for each small plot using community parasite load, calculated as the mean leaf area damaged by all parasites on a host, multiplied by the relative abundance of that host species from the July vegetation survey, and then summed across all hosts in the plot (*Halliday et al., 2019*, *Halliday et al., 2017*; e.g., *Mitchell et al., 2002*).

## Quantification of environmental conditions

Soil temperature (6 cm below the soil surface), soil surface temperature, air temperature (12 cm above the soil surface), and soil volumetric moisture content were recorded at 15 minute intervals for 22–37 days (average 31 days) in the central large plot of each site (n = 22) using a TOMST-4 datalogger (*Wild et al., 2019*). The total duration of measurement varied because some of the dataloggers had to be moved earlier or temporarily because of mowing or grazing activities (*Supplementary file 1a*).

## Statistical analysis

All statistical analyses were performed in R version 3.5.2 (*R Development Core Team, 2015*). We assessed the association between elevation and environmental conditions by fitting linear models with the lm function. All other analyses consisted of fitting linear mixed models with an identity link and Gaussian likelihoods using the lme function in the nlme package (*Pinheiro et al., 2016*). In order to meet assumptions of normality and homoscedasticity, we square-root transformed community parasite load and added an identity variance structure (varIdent function) for each site, which based on visual inspection of residuals of each model, exhibited considerable heteroscedasticity (*Pinheiro et al., 2016*; *Zuur et al., 2009*). Each model included large plots, sites, and meadows as nested random intercepts to account for non-independence among observations due to the sampling design of the CBO. Full equations and parameters for these models are available on Github (https://github.com/fhalliday/Calanda19/tree/Calanda19_disease_submission; *Halliday, 2021*; copy archived at swh:1:rev:86ce01777c396840455fd67a3ff5cd8420e8df21).

We first explored the relationship between elevation and environmental conditions by constructing four models, each including one environmental factor (either mean soil temperature, soil surface temperature, air temperature, or soil moisture) as a response variable, and mean elevation of the site as the predictor.

Next, we explored the relationship between each measure of host community structure (i.e. host species richness and host community pace-of-life) and environmental conditions by constructing two models, each consisting of one measure of host community structure as a response variable and one measure of soil-surface temperature and soil moisture as fixed effects. We only included a single measure of temperature in these models, and excluded elevation, to avoid problems associated with collinearity. We used soil-surface temperature, as this measurement represented the temperature that the majority of leaves (and therefore foliar pathogens) were exposed to (*Figure 2—figure supplement 1*). Results using soil temperature, air temperature, and elevation are reported in the Supplement.

We then tested whether the relationship between host community structure and disease would change as a function of environmental conditions by constructing a mixed model with square-root transformed community parasite load as the response, and soil-surface temperature, soil moisture, host community species richness, and pace-of-life as fixed effects. To estimate whether the effect of host community structure depends on environmental conditions, we also included in the model the pairwise interactions between both measures of host community structure and either soil-surface temperature or soil moisture as additional fixed effects. As before, we only included a single measure of temperature in this model and excluded elevation to avoid problems associated with collinearity. Results using soil temperature, air temperature, and elevation are reported in the Supplement. To aid the interpretation of main effects in the model, we centered all variables so that the mean value of each variable was used as the reference value for interpreting the other variables' independent effects. To evaluate model fit, we calculated the root-mean-squared error (RMSE) of the model, the marginal and conditional pseudo-$R^2$ of the model using the r.squaredGLMM function in the MuMIn package (*Bartoń, 2018*), and the RMSE using leave-one-out cross validation (LOOCV RMSE).

To test whether effects driven by host community pace-of-life were influenced by one or a few important functional traits, we repeated this analysis, including the community-weighted-mean of each leaf trait (leaf chlorophyll content, leaf lifespan, leaf nitrogen content, leaf phosphorus content, and specific leaf area) replacing host community pace-of-life. Individual leaf traits were measured using different units, and were therefore transformed to a common scale using a z-transformation. None of the models including individual leaf traits were improvements over the model including only host community pace-of-life (*Supplementary file 1f*); thus, individual leaf traits were excluded from further analyses.

Finally, to compare direct and indirect effects of environmental conditions on disease risk, we performed confirmatory path analysis using the PiecewiseSEM package (*Lefcheck, 2016*). Specifically, we fit a structural equation model (SEM) that included the effect of elevation on soil-surface temperature, the effect of soil-surface temperature on square-root-transformed disease, the effect of soil-surface temperature on two endogenous mediators (host community species richness and pace-of-life), which together measure changes in host community structure (following *Halliday et al., 2020a*; *Halliday et al., 2019*), and the effects of those two mediators on square-root-transformed community parasite load. We also tested the hypothesis that soil-surface temperature altered the

relationship between host community structure and disease by fitting a second-stage moderated mediation (*Hayes, 2015*) including the pairwise interaction between soil-surface temperature and community pace-of-life, omitting other potential interactions that were non-significant in the model testing whether effects of community structure on disease depend on environmental conditions. Soil moisture was excluded from the SEM because it was unrelated to all other variables in the model. To aid the interpretation of direct effects in the model, we mean-centered soil-surface temperature and host community pace-of-life, so that average soil-surface temperature and host community pace-of-life were used as the reference values for interpreting the other variable's independent effects. We then explored the interaction between community pace-of-life and temperature by setting the reference temperature to one standard deviation above and below the mean temperature, and re-running the model.

## Acknowledgements

We are grateful for insightful suggestions and field assistance from J Alexander, M Maechler, K Raveala, M Tiusanen, A Norberg, I Kohonen, V Loaiza, J Moser, and members of the Laine Lab. This work was supported by Gemeinde Haldenstein, Gemeinde Chur, the University of Zürich, and by grants from the Academy of Finland (296686) to A-LL and the European Research Council (Consolidator Grant RESISTANCE 724508) to A-LL.

## Additional information

### Funding

| Funder | Grant reference number | Author |
|---|---|---|
| Academy of Finland | 296686 | Anna-Liisa Laine |
| European Research Council | 724508 | Anna-Liisa Laine |

The funders had no role in study design, data collection and interpretation, or the decision to submit the work for publication.

### Author contributions

Fletcher W Halliday, Conceptualization, Data curation, Formal analysis, Investigation, Methodology, Project administration, Supervision, Visualization, Writing – original draft, Writing – review and editing; Mikko Jalo, Conceptualization, Data curation, Investigation, Methodology, Project administration, Visualization, Writing – original draft, Writing – review and editing; Anna-Liisa Laine, Conceptualization, funding-acquisition, Project administration, Supervision, Writing – review and editing

### Author ORCIDs

Fletcher W Halliday  http://orcid.org/0000-0003-3953-0861
Anna-Liisa Laine  http://orcid.org/0000-0002-0703-5850

### Decision letter and Author response

Decision letter https://doi.org/10.7554/eLife.67340.sa1
Author response https://doi.org/10.7554/eLife.67340.sa2

## Additional files

### Supplementary files

• Supplementary file 1. Supplementary Tables. (a) Timing of grazing, vegetation and disease surveys and temperature measurements at each site. Recovery days represents the amount of time between the end of grazing activities and the beginning of the vegetation survey at each site. (b) Results of Type II Analysis of Deviance tests on models quantifying the effects of soil moisture and either soil-surface temperature, soil temperature, air temperature, or elevation on two measures of host community structure (Host Richness, Host Pace-of-Life). (c) Results of Type II Analysis of

Deviance test on mixed models of disease, using soil temperature, air temperature, or elevation to evaluate factors that influenced square-root transformed community parasite load. (d) Coefficient estimates from the structural equation model fit with mean-centered soil-surface temperature and host pace-of-life. Estimates are provided both raw (Estimate) and scaled by the ratio of the standard deviation of x divided by the standard deviation of y (Std Estimate) to facilitate comparisons. Correlations among dependent variables are indicated by ~~. (e) Calanda Biodiversity Observatory Vegetation list. This list includes species that were observed during the vegetation survey as well as taxa observed outside of the plots during extensive preliminary surveys of Mount Calanda. (f) Comparison of different models quantifying the relationship between host community traits and disease. Each model contained square-root transformed community parasite load as the response, and elevation, host community species richness, richness-independent phylogenetic diversity, and some combination of host traits as fixed effects. To estimate whether the effect of host community structure depends on elevation, we also included in the model the pairwise interactions between each measure of host community structure and elevation as additional fixed effects, The Pace-of-Life model includes host community pace-of-life as a latent factor, and is the model reported in the manuscript. The All Traits model includes all single traits in place of the pace-of-life latent factor. The Chlorophyll, Leaf Longevity, Leaf Nitrogen, Leaf Phosphorus, and Specific Leaf Area models include a single trait in place of the latent factor.

• Transparent reporting form

### Data availability

The data and code supporting the results are available on Figshare (DOI: 10.6084/m9.figshare.14058059) and Github (https://github.com/fhalliday/Calanda19/tree/Calanda19_disease_submission; copy archived at https://archive.softwareheritage.org/swh:1:rev:86ce01777c396840455fd67a3ff5cd8420e8df21).

The following dataset was generated:

| Author(s) | Year | Dataset title | Dataset URL | Database and Identifier |
|---|---|---|---|---|
| Halliday FW, Jalo M, Laina A-L | 2021 | | https://figshare.com/s/58890c7784189f6c264d | figshare, 10.6084/m9.figshare.14058059 |

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
