## [Decision Letter]

**Acceptance summary:**

This paper provides a framework to disentangle the direct vs. indirect effects of environmental conditions on disease. The authors validate this framework with a well-designed field survey on plant leaf disease across a large elevational gradient. Their results suggest that environmental gradients can alter how host community structure affect disease.

**Decision letter after peer review:**

Thank you for submitting your article "The effect of host community functional traits on disease risk varies along an elevational gradient" for consideration by *eLife*. Your article has been reviewed by 2 peer reviewers, including Yuxin Chen as the Reviewing Editor and Reviewer #1, and the evaluation has been overseen by Detlef Weigel as the Senior Editor.

Essential revisions:

1. Both reviewers find that the current analyses have limited implications for climate change. The revision should either tone down the claim on "global change" or supplement analyses including specific environmental variables in the SEMs (Major #3 of Reviewer 1 and Major #1 and 3 of Reviewer 2).

2. The accurate interpretations of direct and indirect effects from the SEMs would require some clarifications on the modeling processes with interactive terms (Major #1 of Reviewer 1).

3. Please provide a clear indication of the biological system under investigation in your title, e.g,, plant disease.

*Reviewer #1 (Recommendations for the authors):*

1. Direct vs. indirect effects in statistical analyses: the SEM includes interactive terms between elevation and community structures. In this case, the parameters associated with the main effects, elevation and community structures, and their interpretation are dependent on the interactive effects. In other words, you could get different estimates for elevation when you use different reference values of community structures, which applies to the estimates for community structures as well. Under this scenario, it is important to (1) have a clear definition of "direct effect" and "indirect effect" when introducing your analyses and interpreting your results; and (2) state clearly how you did the analyses with interactive terms, e.g., did you do any centering? What reference values did you choose for elevation or community structures? What do you mean by standardizing the estimates to "a common scale" in the legend of Table S6? Are the estimates on the y axis of Figure 3 the raw or standardized coefficients? It seems that the coefficients from Figure 3 are quite different that in Figure 4 and Table S6. The maximum effect of pace-of-life on disease in Figure 3 is 0.4, which is much smaller than "0.66" reported in Figure 4 and Table S6. Is "0.66" based on the reference of elevation=0?

2. Direct vs. indirect effects in the introduction: is the interactive effect between environment and community structure just one kind of indirect effects, or a third category different from direct or indirect effects? It got me confused as you used both definitions in this paper: the legend of Figure 1b vs. the paragraph starting from line 125. Based on the technical report "Direct and Indirect Effects" by Judea Pearl in 2001, I would vote for the definition as the legend of Figure 1b. That is, path d Figure 1b belongs to indirect effects as paths b and c.

3. Global change vs. elevation: the main results are almost based on the analyses with elevation, not climatic variables, which gives very limited implications to global (climate) change. I suggest two alternatives: (1) tone down the expressions on global (climate) change throughout the paper; or (2) supplement more analyses using specific climatic or environmental variables and show more direct implications to climate or environmental change. I saw that you found a strong correlation between elevation and temperature, and a weak correlation between elevation and soil moisture. But this doesn't mean that you can equal the effects of elevation with those of temperature. One simple way to address this issue could be replacing elevation with temperature in your LMMs and SEMs. Or in a more complicated way, add both elevation and environmental variables in SEMs.

*Reviewer #2 (Recommendations for the authors):*

I have substantial concerns that authors may address to improve the manuscript. Please see below.

1. Framework. You hypothesized that elevation can affect disease risk directly by environment factors (e.g., temperature) (line 126-127). And, you did measure temperature along altitude (line 266-269), but you did not add temperature (or other environment factors) in the SEM (Figure 4). You argued that elevation can directly affect disease risk. In fact, elevation is a complex of temperature, humidity, host community, soils and many other factors. Elevation can only influence disease risk by changing biotic and abiotic factors, and the elevation itself is just a series of numbers in SEM. In figure 4A, the "direct" path from elevation to disease risk does not mean the direct effect of elevation on diseases, this refers to those factors that are actually affected by elevation, but are not captured by host community structure, e.g., soils and temperature. So I suggest you include temperature in the SEM, elevation can affect disease risk: (i) by changing the temperature; (ii) by shifting community structure (indirectly); (iii) by altering the relationship between host competence/ pace-of-life and disease risk; (iv) through those factors that are actually affected by elevation, but are not shown in SEM (i.e., the "direct" effect from elevation to disease risk).

2. Is the "trait-competence relationship" (Figure 1A) equal to "relationship between host traits and host competence" (line 130-131), or "the relationship between host community pace-of-life and infection severity (line 34)", or "interaction effect between elevation and host pace-of-life on disease risk" (Table S5 and Figure 3)? Please explain their differences or use consistent terminology. This is the core of the manuscript, please explain carefully.

3. I suggest that the authors weaken the claim of "global change" throughout the manuscript. Global change refers to rising of carbon dioxide, warming, changes in precipitation pattern and many other planetary-scale changes. While elevation is a complex of various environmental factors including temperature, humidity, light, wind, soil, etc. We can infer the potential impact of climate change on disease risk through this study. But, many of the statements in this manuscript make me feel that this study was directly test the impact of climate change on the relationship between community structure and disease risk based on manipulate experiments. E.g., line 28-29, 126, 505-507. In addition, I suggest using "climate change" rather than global change (see Descombes et al. 2020, Science, 370:1469-1473).

---

## [Author Response]

Essential revisions:1. Both reviewers find that the current analyses have limited implications for climate change. The revision should either tone down the claim on "global change" or supplement analyses including specific environmental variables in the SEMs (Major #3 of Reviewer 1 and Major #1 and 3 of Reviewer 2).

Following reviewer comments, we have completely re-analyzed the data, focusing more explicitly on environmental variables. The new results show that our previous results could be entirely attributed to changing temperature associated with elevation. Nevertheless, we have also toned down our claims regarding implications of these results for global change.

2. The accurate interpretations of direct and indirect effects from the SEMs would require some clarifications on the modeling processes with interactive terms (Major #1 of Reviewer 1).

We clarified how we modeled and interpreted the direct and indirect effects in the SEMs, particularly with respect to statistical interactions in the Methods and Results. We further edited Figure 4 (the SEM) to more clearly show the statistical interaction in the model. To do this, we first mean-centered all variables that were influenced by statistical interactions, and then probed the independent effects of these variables by setting the reference values to one standard-deviation above, and below the mean and re-analyzing the data.

3. Please provide a clear indication of the biological system under investigation in your title, e.g,, plant disease.

We have changed the title to, “The effect of host community functional traits on plant disease risk varies along an elevational gradient”.

Reviewer #1 (Recommendations for the authors):1. Direct vs. indirect effects in statistical analyses: the SEM includes interactive terms between elevation and community structures. In this case, the parameters associated with the main effects, elevation and community structures, and their interpretation are dependent on the interactive effects. In other words, you could get different estimates for elevation when you use different reference values of community structures, which applies to the estimates for community structures as well. Under this scenario, it is important to (1) have a clear definition of "direct effect" and "indirect effect" when introducing your analyses and interpreting your results; and (2) state clearly how you did the analyses with interactive terms, e.g., did you do any centering? What reference values did you choose for elevation or community structures? What do you mean by standardizing the estimates to "a common scale" in the legend of Table S6? Are the estimates on the y axis of Figure 3 the raw or standardized coefficients? It seems that the coefficients from Figure 3 are quite different that in Figure 4 and Table S6. The maximum effect of pace-of-life on disease in Figure 3 is 0.4, which is much smaller than "0.66" reported in Figure 4 and Table S6. Is "0.66" based on the reference of elevation=0?

We thank the reviewer for pointing out the lack of clarity regarding how we interpretated direct paths in the SEM in the context of statistical interactions. We have made the following revisions to resolve these problems and to clarify the interpretation of the model-estimated effects.

(1) We now clearly define what is meant by direct and indirect effect throughout the Introduction. To do this, we now refer to indirect effects that are mediated by a shifting endogenous variable (e.g., the indirect effect of temperature on disease via shifting species richness) and indirect effects that are moderated by a third variable (e.g., the effect of community pace-of-life that is moderated by shifting temperature).

(2) We now clearly describe in the Methods what we used as reference values when interpreting direct and indirect effects in both the liner mixed model and the SEM. Specifically, at Line 618 of the manuscript in tracked changes (referring to the linear mixed model), we now say, “To aid the interpretation of main effects in the model, we centered all variables so that the mean value of each variable was used as the reference value for interpreting the other variables' independent effects.” At Line 650 (referring to the SEM), we now say, “To aid the interpretation of direct effects in the model, we mean-centered soil-surface temperature and host community pace-of-life, so that average soil-surface temperature and host community pace-of-life were used as the reference values for interpreting the other variable’s independent effects. We then explored the interaction between community pace-of-life and temperature by setting the reference temperature to one standard deviation above and below the mean temperature, and re-running the model.”

(3) We re-drew the SEM in Figure 4 to show estimates of community-weighted mean host pace-of-life at both low and high reference temperatures. We further clarified the language in the Figure 4 legend to state that, “The High and Low Temperature coefficients are estimated with the reference temperature set to one standard deviation above and below the mean temperature, respectively. All other coefficients are estimated from a model using mean-centered values for temperature and community pace-of-life.”

(4) We added to the Table S6 (Now “Supplementary file 1d”) legend that standardized estimates were scaled by the ratio of the standard deviation of x divided by the standard deviation of y.

(5) We clarified in the Figure 3 legend that coefficients in this figure are raw coefficients from the linear mixed model testing effects of environmental conditions, community structure, and their interaction on disease.

(6) The estimates in Table S6 (Now “Supplementary file 1d”) are derived using mean-centered temperature and pace-of-life. These estimates are comparable to the center of Figure 3. The mean value in Figure 3 is roughly 0.1, which corresponds to the raw estimate of host pace-of-life on disease in Supplementary file 1d (0.1289). We should note that this differs from the values presented in Figure 4, because Figure 4 shows standardized estimates to allow for comparison between different pathways in the model, while Figure 3 shows raw coefficients, and this is clarified on legends for Figure 3 and Figure 4.

2. Direct vs. indirect effects in the introduction: is the interactive effect between environment and community structure just one kind of indirect effects, or a third category different from direct or indirect effects? It got me confused as you used both definitions in this paper: the legend of Figure 1b vs. the paragraph starting from line 125. Based on the technical report "Direct and Indirect Effects" by Judea Pearl in 2001, I would vote for the definition as the legend of Figure 1b. That is, path d Figure 1b belongs to indirect effects as paths b and c.

We agree that the language was confusing, and have edited the Introduction, so that these are now both described as “indirect” effects. We have further clarified the distinction throughout the manuscript by explicitly comparing effects that are described in SEM as mediation (“a” affects “b” which affects “c”) and effects that are a consequence of what is commonly described in SEM as moderation (a statistical interaction: “a” explains under what conditions “b” is related to “c”; e.g., Barron and Kenny 1986 Journal of Personality and Social Psychology).

3. Global change vs. elevation: the main results are almost based on the analyses with elevation, not climatic variables, which gives very limited implications to global (climate) change. I suggest two alternatives: (1) tone down the expressions on global (climate) change through out the paper; or (2) supplement more analyses using specific climatic or environmental variables and show more direct implications to climate or environmental change. I saw that you found a strong correlation between elevation and temperature, and a weak correlation between elevation and soil moisture. But this doesn't mean that you can equal the effects of elevation with those of temperature. One simple way to address this issue could be replacing elevation with temperature in your LMMs and SEMs. Or in a more complicated way, add both elevation and environmental variables in SEMs.

In response to this comment, we both toned down our expression of global change throughout the introduction and added new analyses using specific environmental variables. Specifically, we replaced elevation with temperature and soil moisture in our LMMs, and added both elevation and temperature to the SEM to show that the effects that we previously attributed to changing elevation could also be attributed to changing temperature. We still include models of elevation in the supplement (Supplementary File 1b, Supplementary File 1c), and discuss the elevation results briefly in the Results.

Because temperature and elevation were so tightly correlated (R^2^ > 0.8 for all temperature variables), we also added a brief discussion at Line 357 encouraging readers to interpret the results of the SEM with caution, since the temperature variable might also covary with other factors associated with elevation that were not included in the model, such as changing nutrient concentrations or humidity.

Reviewer #2 (Recommendations for the authors):I have substantial concerns that authors may address to improve the manuscript. Please see below.1. Framework. You hypothesized that elevation can affect disease risk directly by environment factors (e.g., temperature) (line 126-127). And, you did measure temperature along altitude (line 266-269), but you did not add temperature (or other environment factors) in the SEM (Figure 4). You argued that elevation can directly affect disease risk. In fact, elevation is a complex of temperature, humidity, host community, soils and many other factors. Elevation can only influence disease risk by changing biotic and abiotic factors, and the elevation itself is just a series of numbers in SEM. In figure 4A, the "direct" path from elevation to disease risk does not mean the direct effect of elevation on diseases, this refers to those factors that are actually affected by elevation, but are not captured by host community structure, e.g., soils and temperature. So I suggest you include temperature in the SEM, elevation can affect disease risk: (i) by changing the temperature; (ii) by shifting community structure (indirectly); (iii) by altering the relationship between host competence/ pace-of-life and disease risk; (iv) through those factors that are actually affected by elevation, but are not shown in SEM (i.e., the "direct" effect from elevation to disease risk).

In the previous version, we aimed to use elevation as an “umbrella” variable to account for changing conditions along the elevation gradient. However, we see how this use is inconsistent with the idea of direct effects, especially since we did measure relevant abiotic conditions along our gradient, which could actually directly impact disease. We have therefore revised the analyses to more directly test our hypotheses regarding environmental conditions as direct and indirect drivers of disease. To do this, we replaced elevation with temperature and soil moisture in our LMMs. The analyses including Elevation, which were qualitatively similar, are now presented in the Supplement. Following this reviewer comment (and comments from Reviewer 1), we also incorporated temperature into the SEM. Our results show that elevation drives a strong and consistent gradient in temperature, which in turn, alters disease through the three pathways that we previously attributed to elevation.

We also note at Line 357 in the manuscript with tracked changes that because elevation and temperature were strongly correlated, we cannot rule out the possibility that effects attributed to temperature in our model were not driven by other factors associated with elevation, such as nutrient availability or humidity, and we discuss the implications of this when interpreting these results.

2. Is the "trait-competence relationship" (Figure 1A) equal to "relationship between host traits and host competence" (line 130-131), or "the relationship between host community pace-of-life and infection severity (line 34)", or "interaction effect between elevation and host pace-of-life on disease risk" (Table S5 and Figure 3)? Please explain their differences or use consistent terminology. This is the core of the manuscript, please explain carefully.

Thank you for pointing out this source of confusion in the text. We have edited the text to clarify that these are all descriptions of the same phenomenon.

3. I suggest that the authors weaken the claim of "global change" throughout the manuscript. Global change refers to rising of carbon dioxide, warming, changes in precipitation pattern and many other planetary-scale changes. While elevation is a complex of various environmental factors including temperature, humidity, light, wind, soil, etc. We can infer the potential impact of climate change on disease risk through this study. But, many of the statements in this manuscript make me feel that this study was directly test the impact of climate change on the relationship between community structure and disease risk based on manipulate experiments. E.g., line 28-29, 126, 505-507. In addition, I suggest using "climate change" rather than global change (see Descombes et al. 2020, Science, 370:1469-1473).

We thank the reviewer for bringing up an important point. To address this, we revised the text throughout the manuscript to focus more explicitly on shifting environmental conditions associated with climate change, such as warming environmental temperatures, and removed the term “global change” from the manuscript. We further toned down the language throughout the manuscript to place less emphasis on climate change.